# Risk assessment and clinical implications of COVID-19 in multiple myeloma patients: A systematic review and meta-analysis

Sultan Mahmud[1]*, Md. Faruk Hossain[2], Abdul Muyeed[4], Shaila Nazneen[3], Md. Ashraful Haque[5], Harun Mazumder[2], Md Mohsin[3]

**1** Maternal and Child Health Division, International Centre for Diarrhoeal Disease Research, Bangladesh (icddr,b), Dhaka, Bangladesh, **2** Institute of Statistical Research and Training, University of Dhaka, Dhaka, Bangladesh, **3** University of Texas at El Paso (UTEP), El Paso, TX, United States of America, **4** Department of Statistics, Jatiya Kabi Kazi Nazrul Islam University, Trishal, Mymensingh, Bangladesh, **5** Department of Anthropology, Shahjalal University of Science and Technology, Sylhet, Bangladesh

* smahmud@isrt.ac.bd

## Abstract

### Introduction

Patients with multiple myeloma (MM) face heightened infection susceptibility, particularly severe risks from COVID-19. This study, the first systematic review in its domain, seeks to assess the impacts of COVID-19 on MM patients.

### Method

Adhering to PRISMA guidelines and PROSPERO registration (ID: CRD42023407784), this study conducted an exhaustive literature search from January 1, 2020, to April 12, 2024, using specified search terms in major databases (PubMed, EMBASE, and Web of Science). Quality assessment utilized the JBI Critical checklist, while publication bias was assessed using Egger's test and funnel plot. The leave-one-out sensitivity analyses were performed to assess the robustness of the results by excluding one study at a time to identify studies with a high risk of bias or those that significantly influenced the overall effect size. Data synthesis involved fitting a random-effects model and estimating meta-regression coefficients.

### Results

A total of 14 studies, encompassing a sample size of 3214 yielded pooled estimates indicating a hospitalization rate of 53% (95% CI: 40.81, 65.93) with considerable heterogeneity across studies (I2 = 99%). The ICU admission rate was 17% (95% CI: 11.74, 21.37), also with significant heterogeneity (I2 = 94%). The pooled mortality rate was 22% (95% CI: 15.33, 28.93), showing high heterogeneity (I2 = 97%). The pooled survival rate stood at 78% (95% CI: 71.07, 84.67), again exhibiting substantial heterogeneity (I2 = 97%). Subgroup analysis and meta-regression highlighted that study types, demographic factors, and patient comorbidities significantly contributed to the observed outcome heterogeneity, revealing distinct patterns. Mortality rates increased by 15% for participants with a median

**Data Availability Statement:** All relevant data are within the manuscript and its Supporting Information files

**Funding:** The author(s) received no specific funding for this work.

**Competing interests:** The authors have declared that no competing interests exist.

age above 67 years. ICU admission rates were positively correlated with obesity, with a 20% increase for groups with at least 19% obesity. Mortality rates rose by 33% for the group of patients with at least 19% obesity, while survival rates decreased by 33% in the same group.

## Conclusion

Our meta-analysis sheds light on diverse COVID-19 outcomes in multiple myeloma. Heterogeneity underscores complexities, and study types, demographics, and co-morbidities significantly influence results, emphasizing the nuanced interplay of factors.

## Introduction

Cancer poses a substantial global public health challenge, exerting a significant impact across nations and populations. Multiple myeloma (MM), a hematologic malignancy, is characterized by the clonal proliferation of abnormal plasma cells in the bone marrow, resulting in an intrinsic impairment of both humoral and cellular immunity in affected individuals [1, 2]. Plasma cells play a crucial role in producing antibodies necessary to protect the body from infections. The pathogenesis of multiple myeloma (MM) influences the functionality of the adaptive immune system, leading to a reduction in immunoglobulin secretion. MM is characterized by an overproduction of aberrant plasma cells, which can result in bone loss, kidney problems, and various other complications. Furthermore, individuals with multiple myeloma exhibit impaired innate cellular immunity, rendering them more susceptible to a wide range of bacteria and viruses that significantly disrupt their immune system [3].

Multiple myeloma (MM) also presents a unique challenge during the COVID-19 pandemic due to its impact on the immune system and the necessity for intensive treatment regimens. Individuals with MM face an increased susceptibility to severe COVID-19 infection, often requiring hospitalization, incentive cares and exhibiting a higher risk of mortality [4–6]. Numerous studies have examined the impact of COVID-19 on individuals affected by MM in terms of hospitalization, the requirement for intensive care, and mortality/survival rates across different parts of the world with varying sample sizes [1, 4, 7–12]. The majority of these studies are retrospective and case series studies with small sample sizes. The magnitude of these outcomes may differ due to variations in the characteristics of the study population and study regions. As existing studies have depicted variations in the impact of the COVID-19 pandemic on MM patients worldwide, conducting a systematic review and meta-analysis is reasonable to understand the overall picture of that impact. A systematic review and meta-analysis combine data from multiple studies, increasing the statistical power and generalizability of findings that may not be apparent in individual studies due to small sample sizes or variability [13, 14].

However, to date, there have been very few global studies examining the impact of COVID-19 on patients with multiple myeloma (MM), and none of them have been systematically reviewed. However, few systematic reviews and meta-analyses have been conducted to assess the impact of COVID-19 on hematological cancer patients [15, 16]. Unfortunately, all of these studies are based on a large number of single-patient case reports, making it impossible to draw meaningful inferences from such limited data. Therefore, this study would be the first attempt to assess the impact of COVID-19 on MM patients based on studies involving more than one patient. Our systematic review and meta-analysis also intends to comprehensively assess the risk in terms of hospitalization rate, ICU admission rate, mortality rate, survival rate,

and clinical outcomes experienced by MM patients infected with COVID-19, providing crucial insights for healthcare professionals and policy-makers to allocate resources, make triage decisions, and implement preventive measures effectively. These assessments would serve as indispensable remedies that could guide tailored interventions to mitigate the severity of both MM and COVID-19.

Furthermore, these analyses will identify factors influencing these rates including patients' characteristics, comorbidities, and so on, that influence the hospitalization, ICU admission, mortality, and survival rates. The findings will facilitate evidence-based decision-making, risk stratification, personalized patient management, and the development of targeted preventive measures. This research will bridge significant knowledge gaps and support optimal care strategies for multiple myeloma patients during a pandemic like COVID-19.

## Method

This systematic review and meta-analysis followed the guidelines outlined by the Preferred Reporting Items for Systematic Reviews and Meta-Analyses (PRISMA) (S1 Table) [17]. The study was also registered in PROSPERO, the international prospective register of systematic reviews, with the identification number CRD42023407784.

### Search strategy

To facilitate the execution of this systematic review and meta-analysis, an extensive literature search was undertaken spanning the period between April 15 and 16, 2024. The search was limited to publications released from January 1, 2020 to April 12, 2024. The search strategy involved the utilization of Boolean operators ("and," "or") and Medical Subject Headings (MeSH) terms specific to major databases. The designated search queries were encompassed the following terms: "multiple myeloma," "Multiple Myelomas," "Myelomas, Multiple," "Myelomatosis," "Myelomatoses," "Plasma Cell Myeloma," "Myeloma-Multiples," "COVID-19," "coronavirus," "2019ncov," "sars cov 2," "Wuhan," "severe acute respiratory syndrome coronavirus 2," "SARS-CoV-2," "nCoV disease," "2019-nCoV," and "coronavirus 2019." The primary databases explored include PubMed, EMBASE, and Web of Science (See search strings in S2 Table). In addition, the bibliographies of pertinent review articles and selected papers were examined for potentially relevant studies. Due to limited access to certain databases, the search could not be completed as originally intended. Consequently, data from Scopus and Global Health were not included, which represents a deviation from the registered protocol (See S1 Protocol).

### Study selection

The selection criteria for inclusion in this study comprised articles that reported at least one outcome related to hospitalization rate, ICU admission rate, mortality rate, or survival rate for multiple myeloma patients infected with COVID-19. Specifically, studies investigating the impact of COVID-19 on multiple myeloma patients in terms of these outcomes were considered for analysis. To ensure the integrity and relevance of the study, the following criteria were employed to exclude studies: (a) studies that did not measure the impact of COVID-19; (b) studies that did not present data on at least one of the specified outcomes, including mortality rate, survival rate, hospital admission rate, or ICU admission rate; (c) Studies focusing solely on vaccination; (d) studies involving patients with other types of cancers; (e) publications lacking original data, such as expert opinions, consensus statements, editorials, commentaries; (f) studies published in languages other than English; (g) animal studies; (h) studies that were deemed irrelevant to the topic, duplicate publications, reports, single case reports, guidelines,

papers not published in English, and articles lacking sufficient data, including narrative reviews, meta-analyses, systematic reviews, and studies with unavailable full text.

However, research articles and case series were included in the analysis as they provide valuable empirical data. To ensure a rigorous selection process, a two-stage screening approach was implemented. Initially, the titles and abstracts of the identified studies were screened to assess their potential relevance. In the second step, the full texts of the selected studies were thoroughly reviewed to determine their eligibility for inclusion in the study.

### Risk of bias (quality) assessment

In order to evaluate the quality of studies chosen for inclusion in the systematic review and meta-analysis, the critical appraisal checklist offered by the Joanna Briggs Institute (JBI) was utilized [18, 19]. The employment of JBI Critical Appraisal tools provides a standardized and dependable methodology for assessing the quality of research studies, extensively employed in evidence-based practice. The JBI checklist encompasses various study designs, including case series, cohort studies, and case-control studies. Each JBI checklist is tailored to the specific type of study and covers essential elements of study design, data analysis, and reporting. Specifically, for the cohort study, selected studies were evaluated for similarity of groups, the validity and reliability of exposure measurements, the identification and management of confounding factors, and the adequacy of follow-up procedures. Similarly, for the case series study, we focused on assessing the clarity of the case definition, the detailed and systematic reporting of cases, and the consistency in outcome measurement.

Each item on the checklist is scored as '1 = Yes', '0 = No/Unclear', or 'NA = Not Applicable', reflecting the presence or absence of key quality criteria. The scoring was done by two independent reviewers (AH, FH) using Excel format. The total score for each study varies depending on the number of applicable items, with higher scores indicating higher methodological quality. Studies scoring above 70% are considered to be at low risk of bias, those scoring between 50% and 69% are deemed to be at moderate risk of bias, and those scoring below 50% are categorized as being at high risk of bias [20].

The impact of studies with a high risk of bias was assessed by performing a sensitivity analysis using the Leave-One-Out method. This method involves systematically omitting one study at a time from the meta-analysis and recalculating the overall effect estimate to assess the impact of each individual study on the pooled result.

### Screening and extraction

Two investigators (SM and FH) conducted independent screening and data extraction from selected articles using standardized Excel sheets, which served as the data extraction form, adhering to a predefined protocol. Subsequently, the extracted data from both reviewers underwent a comparative cross-check to detect any inconsistencies. Any disagreements were deliberated upon and resolved through further discussion in the presence of another investigator (HR). The pertinent information extracted from the eligible studies encompassed details: first author and published year; sample size; study region; average median age of study participants; proportion of male participants; patients' clinical features; patients' co-morbidity; COVID-19 treatment; MM treatments; hospitalization rate; ICU admission rate; mortality rate; and survival rate.

### Data analysis

The strategy for data synthesis in this systematic review and meta-analysis involved a comprehensive analysis of the data extracted from the included studies. The analysis was conducted

using appropriate statistical methods, to provide a summary estimate of the impact of COVID-19 on multiple myeloma patients. Firstly, a narrative synthesis was conducted to summarize the findings of the included studies and to identify any patterns or trends across the studies. Secondly, the meta-analyses for hospital admission rate, ICU admission rate, mortality rate, and survival rate were carried out using the statistical software STATA 16. The pooled rates with a 95% confidence interval were estimated. Random-effects models were used because of the between-study heterogeneity. Heterogeneity was assessed by computing both the Q test statistic and I2 values. The level of heterogeneity, represented by I2, can be interpreted through Higgins's index: I2 values of 25%, 50%, and 75% signify low, moderate, and high heterogeneity, respectively [21]. Subgroup analyses and meta-regression were performed to investigate potential sources of heterogeneity. Meta-regression additionally gauged the influence of various study characteristics and participants' co-morbidities. The findings from the meta-analyses were depicted in forest plots. To examine publication bias, appropriate statistical methods like funnel plots or Egger's test were employed.

In subgroup and regression analyses, three dummy variables were created based on sample size, median age, and the proportion of male participants, with the median serving as the cut point for each variable. The first dummy variable, related to sample size, was recoded as "1" if the corresponding study had 58 participants or fewer, and "2" otherwise. The second dummy variable was recoded as "1" if the median age of the participants in the corresponding study was 67 years or younger, and "2" otherwise. The dummy variable related to the proportion of males in the study was recoded as "1" if the corresponding study's sample had 58% or fewer males, and "2" otherwise. Among several co-morbidities, this study selected only four of the most common conditions: hypertension, diabetes, Chronic Kidney Disease (CKD), and obesity from the selected studies. Additionally, four dummy variables were generated based on these co-morbidities, using the median proportion of patients as a cutoff point. For the subgroup analysis, the included studies were categorized based on study design as follows:

- Case Series: Studies that present descriptive analysis of cases with a common characteristic, lacking a comparative group.

- Comparative Cohort Studies: These studies identify a cohort (group) of individuals who share a common characteristic or exposure in the past and then look back to compare outcomes between subgroups within this cohort. They typically include a comparison group and allow for some measure of association between exposure and outcome.

- Descriptive Cohort Studies: These studies analyze existing data without the formal structure of a cohort study. They often analyze data from medical records or databases to identify patterns, outcomes, and associations. Descriptive cohort studies do not involve comparing outcomes between different subgroups within the cohort.

## Results

### Study selection

The article selection process for this systematic review and meta-analysis adhered to the guidelines outlined by the PRISMA Checklist, as illustrated in Fig 1.

Initially, a comprehensive search yielded a total of 3125 articles, with 474 from PubMed, 2069 from EMBASE, and 582 from Web of Science. Among these, 446 articles were deemed irrelevant to the topic under investigation, while 558 articles were identified as not relevant to the topic or duplicates. Subsequently, these irrelevant articles were excluded, resulting in a total of 201 studies proceeding to the first step of screening, which involved evaluating their

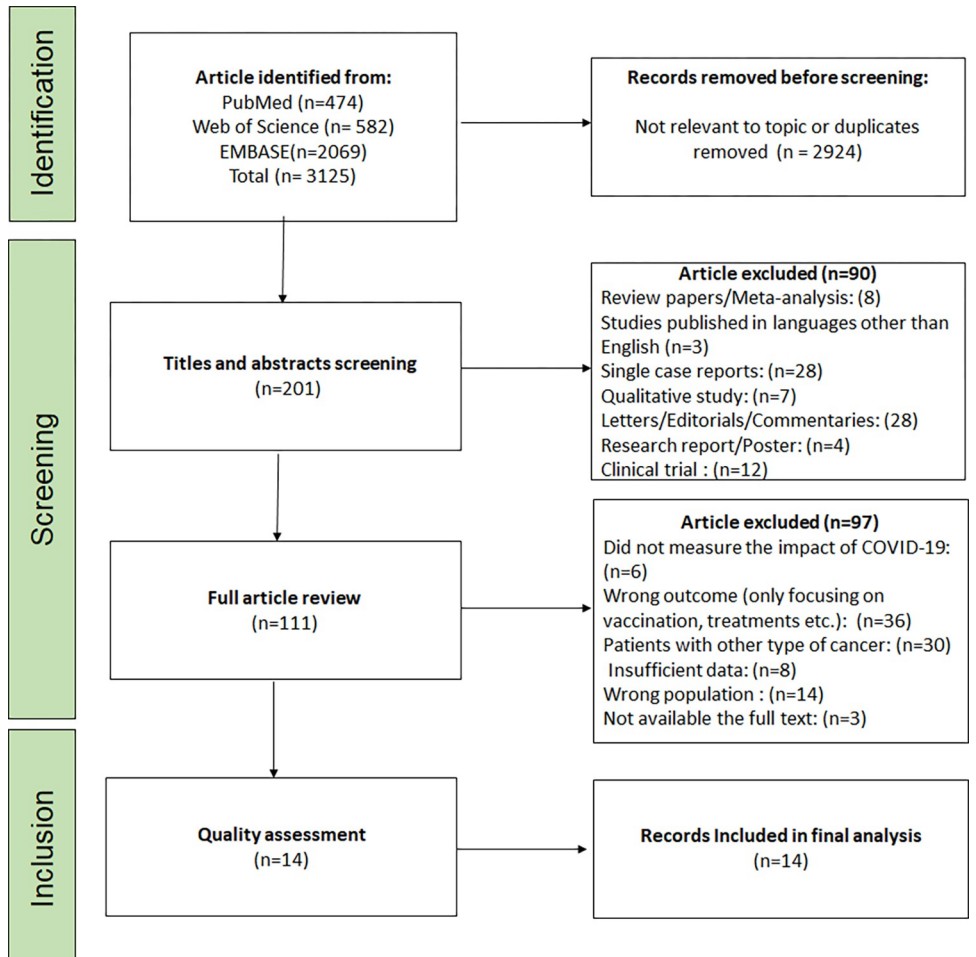

**Fig 1. Flow diagram outlining the search approach and the criteria for including or excluding studies, adhering to the PRISMA-2009 guidelines.**

titles and abstracts. During this initial screening stage, the primary reason for excluding articles was their classification as an irrelevant publication type (90 articles). Following this screening, the full texts of 111 studies were thoroughly reviewed. Ultimately, 14 studies were included in the analysis for further assessment of publication bias or quality. The studies that were excluded during the full-text review process had various reasons for exclusion, including did not measure the impact of covid-19 (6), wrong outcome measured (36), patients with other type of cancers (30), insufficient data (8), wrong population (14), and unavailability of the full text (3). After conducting the quality assessment, all 14 studies that remained were deemed suitable for the final analysis, taking into account their methodological rigor and adherence to the predetermined criteria.

## Study characteristics

The characteristics of the selected studies [1, 4–8, 11, 12, 22–27] are presented in Table 1. Out of the 14 studies that were selected, 12 were single-country studies, while one study was conducted across four distinct countries and another one in 32 European countries. Specifically, the United States accounted for a total of four studies, followed by Sweden with two studies,

**Table 1. Summary characteristics of the selected studies.**

| Author | Published year | Country | Type of study | Study period | Sample size | Median age | Male (%) | Hospitalization rate (95% CI) | ICU admission rate (95% CI) | Mortality rate (95% CI) | Survival rate (95% CI) | JBI Score; Risk of bias |
|---|---|---|---|---|---|---|---|---|---|---|---|---|
| Wang et al. | 2020 | USA | Descriptive cohort studies | March 1, 2020-April 30, 2020 | 58 | 67 | 52.0 | 62 (49.51, 74.49) | 30 (18.21, 41.79) | 39 (26.45, 51.55) | 61 (48.45, 73.55) | 8; Low |
| Chari et al. | 2020 | Spain, France, USA, UK | Descriptive cohort studies | January 2019-December 2020 | 650 | 69 | 58.5 | 68.77 (65.21, 72.33) | 14 (11.33, 16.67) | 34.1 (30.46, 37.74) | 65.9 (62.26, 69.54) | 7; Moderate |
| Hultcrantz et al. | 2020 | USA | Descriptive cohort studies | March 1, and April 30, 2020 | 100 | 68 | 58.0 | 74 (65.4, 82.6) | 16 (8.81, 23.19) | 24 (15.63, 32.37) | 76 (67.63, 84.37) | 8; Low |
| Ho et al. | 2022 | USA | Descriptive cohort studies | December 1, 2019- August 31, 2021 | 187 | 65 | 56.0 | 39 (32.01, 45.99) | 10 (5.7, 14.3) | 5 (1.88, 8.12) | 95 (91.88, 98.12) | 8; Low |
| Martínez-López et al. | 2020 | Spain | Case-series study | March 1, 2020-April 30, 2020 | 167 | 71 | 57.0 | 99.99 (99.84, 100.14) | 21 (14.82, 27.18) | 34 (26.82, 41.18) | 66 (58.82, 73.18) | 10; Low |
| Krejci et al. | 2021 | Czech Republic | Descriptive cohort studies | October 2020-February 2021 | 50 | 68 | 64.0 | 56 (42.24, 69.76) | 20 (8.91, 31.09) | 18 (7.35, 28.65) | 82 (71.35, 92.65) | 8; Low |
| Susek et al. | 2020 | Sweden | Case-series study | March 2020- May 2020 | 9 | 70.4 | 66.0 | 44 (11.57, 76.43) | (0, 0) | 44 (11.57, 76.43) | 56 (23.57, 88.43) | 9; Low |
| Ehsan et al. | 2023 | USA | Comparative Cohort Studies | March 1, 2020-October 30, 2020 | 162 | 64 | 57.0 | 20.3 (14.11, 26.49) | 9.8 (5.22, 14.38) | 6 (2.34, 9.66) | 94 (90.34, 97.66) | 9; Low |
| Silfverberg et al. | 2022 | Sweden | Comparative Cohort Studies | January 1, 2020-December 31, 2020 | 20 | 60 | 55.0 | 30 (9.92, 50.08) | 10 (-3.15, 23.15) | 10 (-3.15, 23.15) | 90 (76.85, 103.15) | 7; Moderate |
| Jin et al. | 2023 | China | Comparative Cohort Studies | January 1, 2022-December 31, 2022 | 509 | | 59.0 | 15.1 (11.99, 18.21) | 3.3 (1.75, 4.85) | 3.1 (1.59, 4.61) | 96.9 (95.39, 98.41) | 8; Low |
| Karadeniz et al. | 2022 | Turkey | Descriptive cohort studies | April 1, 2020 -March 30, 2021 | 30 | 63 | 67 | 63 (45.72, 80.28) | 27 (11.11, 42.89) | 30 (13.6, 46.4) | 70 (53.6, 86.4) | 7; Moderate |
| Musto et al., | 2023 | 32 European countries | Comparative Cohort Studies | February 2020-August 2022 | 1221 | 68 | 58.0 | 64 (61.31, 66.69) | 14 (12.05, 15.95) | 25 (22.57, 27.43) | 75 (72.57, 77.43) | 8; Low |
| Radoch et al. | 2021 | Czech Republic | Descriptive cohort studies | March 2020-January 2021 | 158 | 70 | 50.0 | 40 (32.36, 47.64) | 17 (11.14, 22.86) | 27 (20.08, 33.92) | 73 (66.08, 79.92) | 6; Moderate |
| Garnica et al., | 2023 | Brazil | Comparative Cohort Studies | April 2020—January 2022 | 91 | 62 | 50 | 66 (56.27, 75.73) | 37 (27.08, 46.92) | 30 (20.58, 39.42) | 70 (60.58, 79.42) | 8; Low |

and the Czech Republic with two studies. Meanwhile, a single study took place in Brazil, Turkey, Spain, and China respectively. The average sample size was 243 (ranging from 9 to 1221) with a cumulative sample size of 3214 participants. The average median age was 67 (range: 60–71). It is noteworthy that the median age of the participants was below 68 years in 50% of the selected studies. The average proportion of male participants was 58 (range: 50–67), and the male gender constituted more than 59% of the participants in 50% of the studies.

## Patients' clinical features, co-morbidity, follow-up time, and treatments

Table 2 provides a comprehensive overview of the clinical features and comorbidities in patients with multiple myeloma (MM) who were also diagnosed with COVID-19, as well as the treatment strategies used for both conditions. The most common clinical features included fever, cough, and dyspnea. Fever was reported in 40% to 100% of patients, while cough was observed in 65% to 100%. Dyspnea had a frequency ranging from 32.9% to 45%. Additional reported symptoms included fatigue, sore throat, myalgia, diarrhea, and gastrointestinal symptoms.

Hypertension was the most prevalent comorbidity, affecting 33.3% to 70% of patients in different studies. Diabetes was also a frequent comorbidity, with a prevalence ranging from 16.77% to 44.4%. Other notable comorbidities included chronic kidney disease (CKD), obesity, cardiac diseases, and lung diseases. Some studies also noted hyperlipidemia and peripheral neuropathy.

COVID-19 treatment strategies varied across the studies. The most common treatments included oxygen support, remdesivir, hydroxychloroquine, azithromycin, antibiotics, and dexamethasone. Other treatments, like convalescent plasma, IL-6 blockers, and corticosteroids, were also used in some cases.

For MM treatment, immunomodulatory drugs (IMiDs) were frequently employed, often in combination with other therapies. Proteasome inhibitors and anti-CD38 antibodies were also widely utilized. Other MM treatments included autologous stem cell transplantation (ASCT), daratumumab-based therapy, and monoclonal antibodies.

## Risk of bias and quality assessment

Based on the JBI Critical Appraisal score, a total of 10 studies were assessed as having a low risk of bias, scoring between 8 and 10 (Table 1 and S3 Table). These studies demonstrated a high percentage of positive responses to the questions on the checklist, indicating strong methodological quality. Additionally, 4 studies were identified as having a moderate risk of bias, scoring between 6 and 7. None of the studies fell into the high-risk category, which would have been indicated by scores of 5 or below. This assessment underscores the overall robustness of the included studies.

## Pooled estimates

The analysis revealed that the pooled hospitalization rate among COVID-19 patients with multiple myeloma cancer was 53% (95% confidence interval [CI]: 40.81, 65.93) (Fig 2A). However, there was a significant amount of heterogeneity observed among the selected studies used in this analysis (I2 = 99%, p-value for Q-test <0.001). To assess publication bias, we examined the Funnel plot (Fig 3A) and conducted Egger's test (z = -0.71, p-value = 0.48) (Table 3), both of which indicated no significant publication bias among the selected studies. Moreover, Fig 4A shows that the pooled hospitalization rate remains relatively unchanged regardless of which study is excluded, suggesting an overall robust estimate.

**Table 2. Clinical features, co-morbidities, and treatment of patients with COVID-19 and MM.**

| Study | Clinical feature n (%) | Co-morbidity n (%) | COVID-19 treatment n (%) | MM treatment n (%) |
|---|---|---|---|---|
| Wang et al. | Fever 41 (70%), Cough 38 (65%), Dyspnea 26 (45%) | HTN 37 (64%), Hyperlipidemia 36 (62%), Obesity 21 (36%), Diabetes 16 (28%), Chronic Kidney Disease 14 (24%), Lung Disease 12 (21%), Current or Former Smoker 21 (36%), CAD and/or CVD 13 (22%), Heart Failure 7 (12%) | Oxygen Support 10 (17%), RDV 1 (1.7%), HCQ 17 (29%), AZ 17 (29%), Antibiotics 19 (33%), Corticosteroid 10 (17%), Plasma 1 (1.7%), Selinexor 5 (9%), Anti-IL-6 4 (7%), Anti-IL-1 2 (3.45%), Anti TNF 1 (1.7%) | Daratumumab 28 (48%), Immunomodulatory Drugs 32 (55%), Proteasome Inhibitor 22 (38%), Venetoclax 5 (8.6%), Corticosteroids 30 (51.72%) |
| Chari et al. | NR | NR | Combination Strategies 455 (70%), Antibiotics 91 (14%), Hydroxichloroquine 65 (10%) | PIs 559 (86%), IMiDs 520 (80%), Anti-CD38 Antibody 195 (30%) |
| Hultcrantz et al. | Fever, Cough, Fatigue, Sore Throat, Shortness of Breath, | HTN, DM | Hydroxychloroquine 52 (52%), Azithromycin 52 (52%), Combination Hydroxychloroquine and Azithromycin 40 (40%), IL-6 Blockade 9 (9%), Lopinavir-Ritonavir 4 (4%), Remdisivir 1 (1%), Convalescent Plasma 2 (2%) | High dose Melphalan with ASCT 39 (39%), Bortezomib-including Regimen 20 (20%), Carfilzomib-including Regimen 15 (15%), Daratumumab-including Regimen 24 (24%), Ixazomib-including Regimen 6 (6%), Lenalidomide maintenance 22 (22%) |
| Ho et al. | NR | NR | NR | NR |
| Martínez-López et al. | NR | Cardiac Disease 35 (21%), Pulmonary Disease 23 (13.8%), Diabetes 28 (16.77%), Renal Disease 32 (19.2%), HTN 67 (40.12%) | HCQ 148 (88.6%), AZ 91 (54.5%), Antiretrovirals 103 (61.68%), Steroids 83 (49.7%), Anti-interleukin-6 Receptor Antibody Therapy 22 (13.17%), Heparin 109 (65.3%), Oxygen Support 128 (76.65%) | Proteasome Inhibitor 138 (82.6%), Immunomodulatory Drug 119 (71.3%), Monoclonal Antibody 38 (22.75%) |
| Engelhardt et al. | Cough 17 (81%), Fever 16 (76%), Myalgia 4 (19%), GI Symptoms 2 (9.5%) | Cardiac/ HTN 11 (52%), Renal Impairment 3 (14.2%), Obesity 1 (4.7%), PNP 4 (19%), Diabetes 4 (19%), Hypothyreosis 4 (19%) | Antibiotics 17 (81%), AZ 4 (19%), HCQ 7 (33.3%), RDV 1 (4.7%), Tocilizumab 1 (4.7%), Anakinra 1 (4.7%), Oxygen Support 3 (14.2%) | Daratumumab-combination 5 (23.8%), Elotuzumab-combination 1 (4.7%), VCd/ KRd 2/1, Lenalidomide 3 (14.2%), None 9 (42.9%) |
| Krejci et al. | Fever, Dry Cough, Dyspnea, Hypoxia, Arthralgia, Myalgia, Fatigue, Headaches, Diarrhea | Concomitant Cardiovascular or Pulmonary Comorbidities 96%, HTN 70% | Ambulatory Course 22 (44%), Oxygen Therapy 18 (36%), Remdesivir and Convalescent 12 (24%), Dexamethasone or Methylprednisolone 12 (24%), Antibiotics 4 (8%) | Daratumumab-based Therapy 30%, Bortezomib-based Therapy 24%, Carfilzomib or Pomalidomide 12%, Others 6 (12%) |

(*Continued*)

**Table 2.** (Continued)

| Study | Clinical feature n (%) | Co-morbidity n (%) | COVID-19 treatment n (%) | MM treatment n (%) |
|---|---|---|---|---|
| Susek et al. | Fever 9 (100%), Cough 8 (89%), Dyspnea 3 (33.3%), Diarrhea 3 (33.3%), Arthralgia 3 (33.3%), Ageusia | Diabetes 4 (44.4%), HTN 3 (33.3%), Obesity 2 (22.2%) | Oxygen Support 4 (44.4%) | Daratumumab 6 (66.7%), DEX 8 (88.9%), Venetoclax 1 (11.1%), Carfilzomib 1 (11.1%), Bortezomib 1 (11.1%), Lenalidomide 3 (33.3%) |
| Jimenez-Zepeda et al. | NR | NR | Corticosteroid | IMiD 9 (56.25%), PI 3 (18.75%), Daratumuma-based Regimes 6 (37.5%), Dexamethasone 4 (25%), Bortezomib 2 (12.5%) |
| Ehsan et al. | Fever 65 (40%), Cough 86 (53%), Fatigue 90 (56%), Shortness of Breath 61 (38%), Myalgias 48 (30%) | HTN 72 (44.4%), Hypogammaglobinemia 52 (32%), Diabetes 36 (22.2%), Chronic Kidney Diseasse 48 (29.6%), Obesity 27 (16.6%), Chronic Heart Disease 23 (14.1%), Congestive Heart Failure 23 (14.1%), Coronary Artery Disease 22 (13.5%), Other Heart Disease 26 (16%) | Oxygen support 31 (19%), Plasma, Convalescent, Lenzilumab, IVIg, Tocilizumab, Dexamethasone, Remdesivir | ASCT 77 (47.5%), Immunomodulatores (IMiDs) 57 (35.1%), Proteasome Inhibitors (PIs) 46 (28.3%), Anti-CD38 Antibody 43 (26.5%), Maintenance Therapy 48 (29.6%), Induction Therapy 19 (11.7%) |
| Silfverberg et al. | Fever (100%), Cough (100%), Headache (5%) | Diabetes 2 (10%), HTN 4 (20%), Obesity 3 (15%), Chronic Lung Disease 1 (5%), Chronic Kidney Diseas 1 (5%), Organ Tranplantation (kidney) 1 (5%) | Convalescent Plasma 1 (5%), Oxygen support 4 (20%), Glucocorticoids 4 (20%), Remdesivir 1 (5%) | ASCT 20 (100%), Rituximab 1 (5%), Bendamustine 1 (5%) |
| Jin et al. | Fever 302 (59.3%), Respiratory symptoms (Cough, sneezing, runny nose, sore throat, headaches, muscle aches, shortness of breath, fever) 414 (81.3%), Pneumonia 192 (37.7%) | HTN 118 (23.2%), Diabetes 49 (9.6%), Obesity 109 (21.4%), CKD (Chronic kidney disease) 112 (22%), Cardiac 73 (14.3%), Pulmonary comobidity 35 (6.9%) | NR | ASCT 25 (4.9%), Daratumumab_based_therapy 117 (23%), PI/IMiD 392 (77%), |
| Zheng et al. | NR | HTN 8 (24%), Hyperlipidemia 2 (6%), Diabetes 4 (12%), Heart Failure(CHF/Congestive Heart Failure) 3 (9%), Arthritis 3 (9%), CAD 1 (3%), | NR | Dara 10 (29%), VRD 15 (44%), DEX 11 (32%), KRD 4 (12%), REV 6 (18%), Cytoxan 2 (6%), |
| Karadeniz et al. | NR | NR | NR | AHSCT 19 (63%%) |

(*Continued*)

**Table 2.** (Continued)

| Study | Clinical feature n (%) | Co-morbidity n (%) | COVID-19 treatment n (%) | MM treatment n (%) |
|---|---|---|---|---|
| Musto et al. | Pulmonary 451 (36.9%), Extra Pulmonary 224 (18.4%) | Diabetes 192 (15.7%), Obesity 95 (7.8%), COPD 177 (14.5%), HTN 467 (38.2%) | Antivirals/ Corticosteroids/ Plasma 135 (11.1%), Antivirals +Monoclonal Antibodies/ Corticosteroids/ Plasma 23 (1.9%), Monoclonal Antibodies/ Corticosteroids/ Plasma 84 (6.9%), Corticosteroids/ Plasma 10 (.8%), Corticosteroids 94 (7.7%), No treatment 270 (22.1%), Unknown 605 (49.5%) | Imids 698 (57.2%), Cyclophosphamide/Melphalan 50 (4.1%), Monoclonal Antibodies (Daratumumab, Isatuximab, Elotuzumab) 247 (20.2%), Antibodydrug Coniugate and Antibodies 20 (1.6%), Other 19 (1.6%), No Treatment 86 (7%) |
| Radoch et al. | Fever 78 (49.4%), Cough 64 (40.5%), Dyspnea 52 (32.9%), Headache 16 (10.1%) | NR | Convalescent Plasma 4 (2.5%), Hydroxychloroquine 1 (.6%), Remdesivir 14 (8.8%) | IMids 150 (95%%), PI 142 (89.9%), Anti CD38 29 (18.4%), ASCT 58 (36.7%), |
| Garnica et al. | Fever 55 (61%), High respiratory tract 33 (37%), Low respiratory tract 47 (52%), Dyspnea/ Shortness of Breath 33 (37)% | HTN 42 (52%), Diabetes 19 (23%), Obesity 4 (5%), Cardiac 12 (15%), COPD 6 (7%), Kidney 8 (10%) | NR | Corticosteroids 57 (63%), Immunomodulatory Drugs 51 (55%), Proteasome Inhibitors (PI) 38 (42%), Monoclonals 25 (27%), Alkylating Agents 19 (21%) |

NR: Not reported; HTN, Hypertension; DM, Diabetes Mellitus; CAD, Coronary artery disease; CVD, Cerebrovascular disease; RDV, Remdesivir; AZ, Azithromycin; HCQ, Hydroxychloroquine; IL, Interleukin; TNF, Tumor necrosis factor; PI, Proteasome inhibitor; IMiD, Immunomodulatory drug; ASCT, Autologous Stem Cell Transplant; DEX, Dexamethasone; COPD, Chronic obstructive pulmonary disease; VRD/ REV, Velcade® (bortezomib) + Revlimid® (lenalidomide) + dexamethasone;

In terms of the pooled ICU admission rate, it was found to be 17% (95% CI: 11.74, 21.37) (Fig 2B). Again, a significant level of heterogeneity was observed among the selected studies (I2 = 94%, p-value for Q-test <0.001). Additionally, the Funnel plot (Fig 3B) and Egger's test (z = 3.12, p-value = 0.001) (Table 3) indicated significant publication bias. Consistent ICU admission rates across all iterations in sensitivity analysis (Fig 4B) also indicate a reliable and robust estimate.

Regarding the pooled mortality rate, it was estimated to be 22% (95% CI: 15.33, 28.93) (Fig 2C). The heterogeneity test revealed statistically significant between-study heterogeneity (I2 = 97%, p-value for Q-test <0.001). The Funnel plot (Fig 3C) and Egger's test (z = 1.87, p-value = 0.06) (Table 3) suggested that there was no significant publication bias in the studies

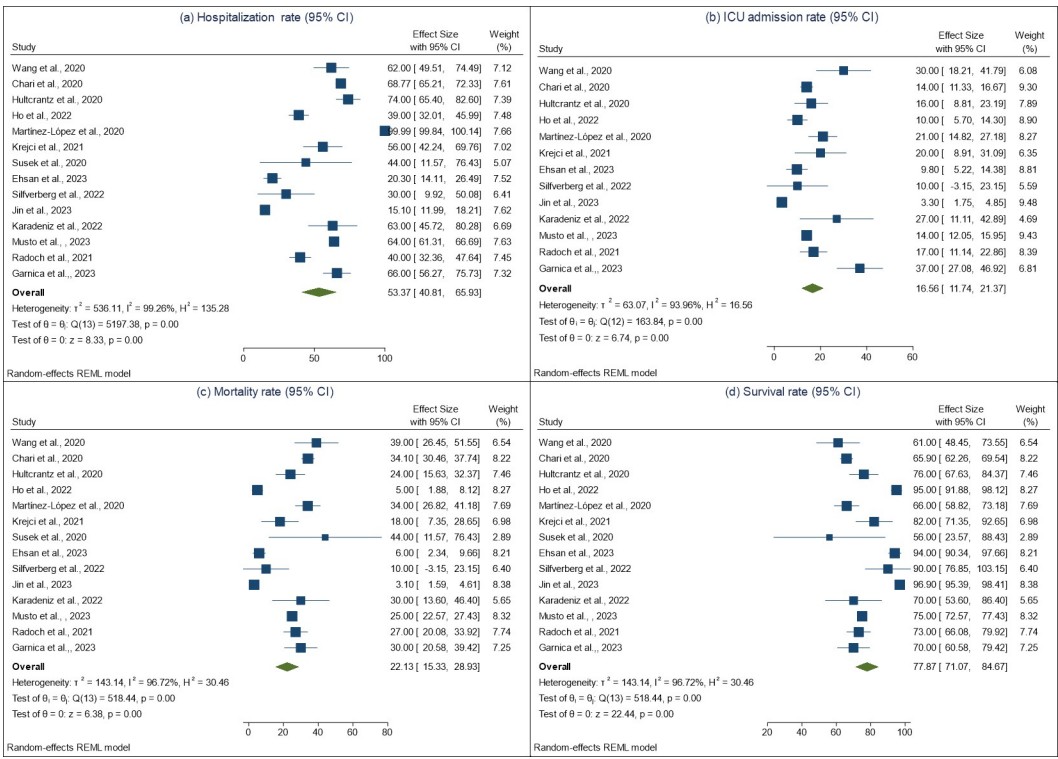

**Fig 2.** Forest plot for (a) hospitalization rate, (b) ICU admission rate, (c) mortality rate, and (d) survival rate among patients with COVID-19 and multiple myeloma based on a random-effects model.

used to estimate the pooled mortality rate. Lastly, the pooled survival rate was determined to be 78% (95% CI: 71.07, 84.67) (Fig 2D). There was a statistically significant level of heterogeneity observed among the studies used for estimating the pooled survival rate (I2 = 97%, p-value for Q-test <0.001). The Funnel plot (Fig 3C) and Egger's test (z = -1.87, p-value = 0.06) (Table 3) indicated no significant publication bias in the selected studies. The sensitivity analysis showed the robustness of the meta-analysis results for both mortality and survival rates (Fig 4C and 4D), suggesting that they are not unduly influenced by any single study.

## Subgroup analysis

**Subgroup analysis of study characteristics and demographic factors.** The subgroup analysis revealed that the type of study significantly contributed to heterogeneity across all four outcomes: hospitalization rate, ICU admission rate, mortality rate, and survival rate. Specifically, variations were observed in the hospitalization rate among different study designs. The rate was 74% (95% CI: 19.78, 129.10; I2 = 91.27%; k = 2) for case- series studies, 39% (95% CI: 17.45, 60.97; I2 = 98.98%; k = 5) for comparative cohort studies, and 57% (95% CI: 46.61, 68.03; I2 = 96.88%; k = 4) for descriptive cohort studies (Fig 5A and S4 Fig 1 in S1 File). No statistically significant differences were observed between these study types concerning hospitalization rates (Q(2) = 2.67; p-value = 0.26). Similarly, the ICU admission rate exhibited variability across different study types, with rates of 21% (95% CI: 14.82, 27.18; k = 1) for case-series studies, 14% (95% CI: 3.41, 25.36; I2 = 98.27%; k = 5) for comparative cohort studies, and 17% (95% CI: 12.40 21.13; I2 = 68.30.%; k = 7) for descriptive cohort studies (Fig 5B and S4 Fig 2 in S1 File). Despite these differences, the between-group variations were not

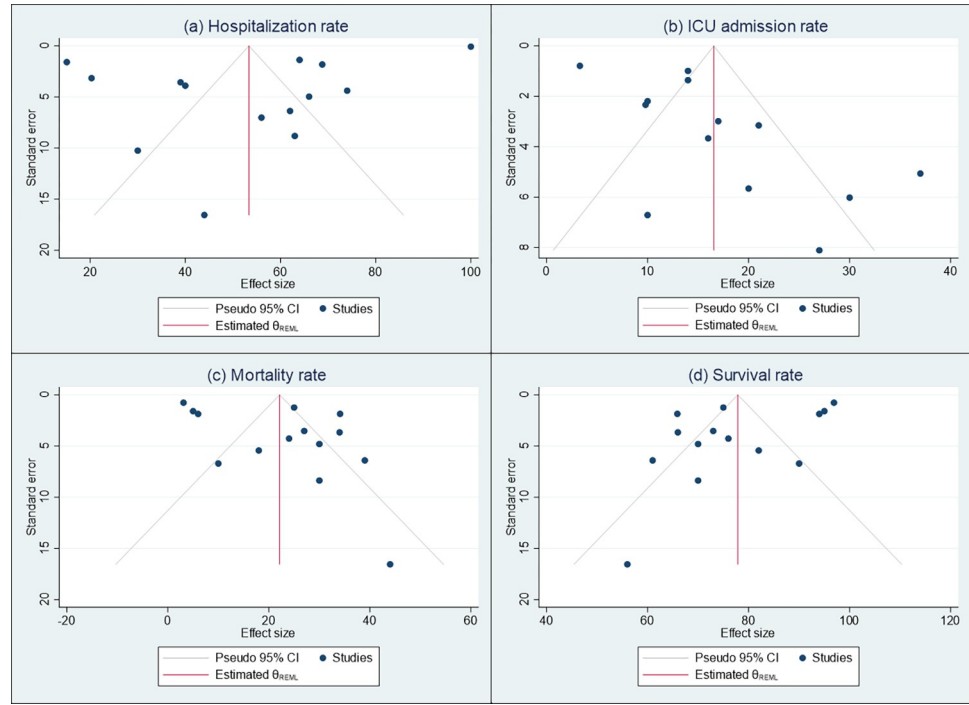

**Fig 3.** Funnel plots presenting the publication bias among selected studies on (a) hospitalization rate, (b) ICU admission rate, (c) mortality rate, and (d) survival rate.

statistically significant (Q(2) = 1.62; p-value = 0.45). On the other hand, the mortality rate exhibited distinct patterns among study types, with rates of 34% (95% CI: 27.45, 41.48; I2 = 0%; k = 2) for case-series studies, 15% (95% CI: 3.97, 25.30; I2 = 97.99%; k = 5) for comparative cohort studies, and 25% (95% CI: 15.81, 33.59; I2 = 92.52%; k = 7) for descriptive cohort studies (Fig 5C and S4 Fig 3 in S1 File). Importantly, the differences between these groups were found to be statistically significant at 5% significant level (Q(2) = 9.76; p-value = 0.01). The survival rate also exhibited distinct patterns among study types, with rates of 66% (95% CI: 58.52, 72.55; I2 = 0%; k = 2) for case-series studies, 85% (95% CI: 74.70, 96.03; I2 = 97.99%; k = 5) for comparative cohort studies, and 75% (95% CI: 66.41, 84.19; I2 = 92.52%; k = 7) for descriptive cohort studies (Fig 5D and S4 Fig 4 in S1 File). The differences between these groups were also found to be statistically significant at 5% significant level (Q(2) = 9.76; p-value = 0.01).

Additionally, when considering the influence of sample size on outcomes, the hospitalization rate was 59% (95% CI: 49.14, 69.33; I2 = 72.50%; k = 7) for studies with 129 or fewer participants, and 50% (95% CI: 27.44, 71.93; I2 = 99.77%; k = 7) for studies with more than 129 participants (Fig 5A and S4 Fig 5 in S1 File). However, the hospitalization rate between these

**Table 3. Egger test results for each of the outcomes.**

| Outcomes | Test Statistic, *Z-value* | *p-value* |
|---|:---:|:---:|
| Hospilalization rate | -0.71 | 0.48 |
| ICU admission are | 3.12 | 0.001 |
| Mortality rate | 1.87 | 0.06 |
| Survival rate | -1.87 | 0.06 |

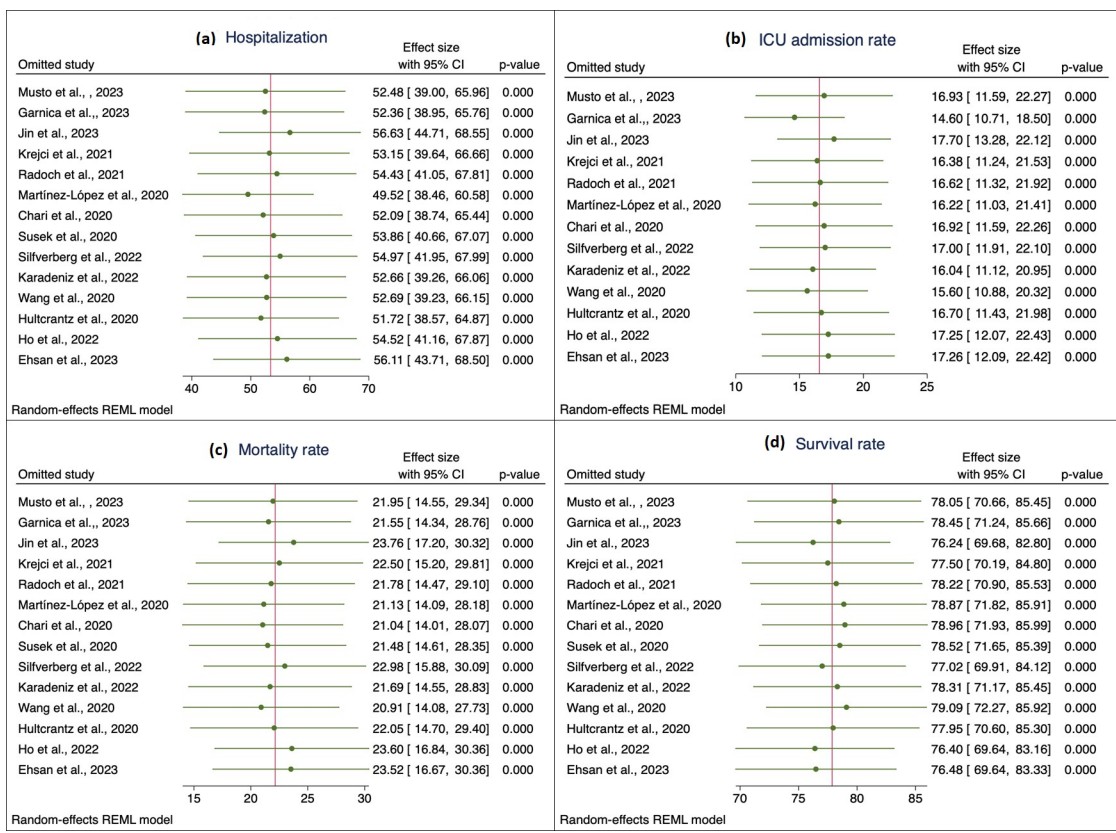

**Fig 4.** Sensitivity analysis exploring the influence of each study on the pooled (a) hospitalization rate, (b) ICU admission rate, (c) mortality rate, and (d) survival rate using the leave-one-out method. Green dot, pooled effect estimate; green horizontal line, confidence interval; red vertical line, pooled effect estimate.

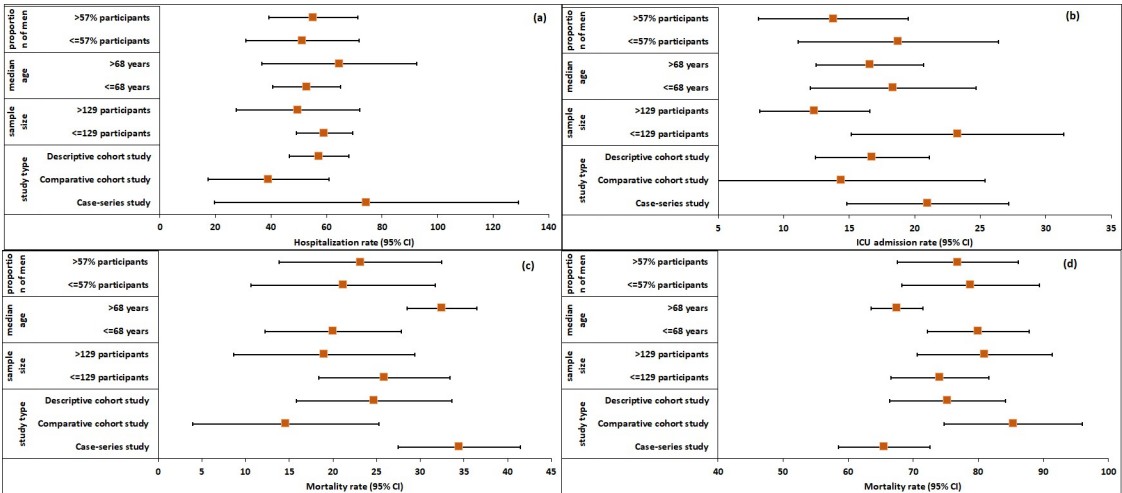

**Fig 5.** Subgroup analysis for (a) hospitalization rate, (b) ICU admission rate, (c) mortality rate, (d) survival rate in patients with COVID-19 and multiple myeloma by different study characteristics.

two groups was not statistically significant (Q(1) = 0.59; p-value = 0.44). The ICU admission rate was 23% (95% CI: 15.16, 31.38; I2 = 69.35%; k = 6) for studies with 129 or fewer participants, and 12% (95% CI: 8.15, 16.58; I2 = 92.62%; k = 7) for studies with more than 129 participants (Fig 5B and S4 Fig 6 in S1 File). The ICU admission rate between these two groups was statistically significant (Q(1) = 5.47; p-value = 0.002). However, there were no statistically significant differences in mortality rate, and survival rate between studies with different sample sizes (S4 Fig 7, 8 in S1 File). Furthermore, when considering the participants' median age, the hospitalization rate, ICU admission rate, mortality rate, and survival rate exhibited variations between participants with a median age of 67 years or less and those with a median age more than 67 years (Fig 5A–5D). The differences in mortality rate, and survival rate were found to be statistically significant at a 5% level of significance (S4 Fig 11, 12 in S1 File), while the differences in ICU admission and hospitalization rates were not statistically significant (S4 Fig 9, 10 in S1 File). Lastly, the proportion of male participants did not significantly influence the hospitalization rate, ICU admission rate, mortality rate, or survival rate.

**Subgroup analysis of patients' co-morbidity.** An analysis of subgroups by the proportion of hypertension (HTN) patients revealed that the pooled estimate of the hospitalization rate among patients with COVID-19 and multiple myeloma was 55% for the group with more than 42% HTN patients and 51% for the group with less than or equal to 42% HTN patients (Table 4). The estimated hospitalization rate was found to be 48% among the group with more than 18% diabetes patients and 57% among the group of patients with less than or equal to 18% diabetes patients. The group with more than 19% obesity patients and the group with less than or equal to 19% obesity patients had ICU admission rates of 39% and 46%, respectively. Additionally, the pooled hospitalization rate for the group with at least 19% chronic kidney disease (CKD) patients was 32%, and for the group with less than or equal to 19% CKD patients, it was 66%. There was no significant difference between the two groups for all co-morbidities considered, except for CKD, which was significant at the 10% level (Q(1) = 2.86; p = 0.09).

**Table 4. Subgroup analysis for hospitalization rate, ICU admission rate, mortality rate, survival rate in patients with COVID-19 and multiple myeloma by co-morbidities.**

| Variable: Co-morbidity | Labels | Hospitalization rate¥ (95% CI) | Q(b)*; P-value | ICU admission rate¥ (95% CI) | Q(b); P-value | Mortality rate¥ (95% CI) | Q(b); P-value | Survival rate¥ (95% CI) | Q(b); P-value |
|---|---|---|---|---|---|---|---|---|---|
| Hypertension | < = 42% participants | 51.28 (20.96, 81.60) | Q(1) = 0.05; p = 0.82 | 11.97 (4.04, 19.89) | Q(1) = 2.37; p = 0.12 | 20.77 (7.19, 34.35) | Q(1) = 0.04; p = 0.83 | 79.23 (65.65, 92.81) | Q(1) = 0.04; p = 0.83 |
| | >42% participants | 55.42 (36.56, 74.28) | | 21.89 (12.07, 31.71) | | 22.66 (11.53, 33.79) | | 77.34 (66.21, 88.47) | |
| Diabetes | < = 18% participants | 57.03 (26.88, 87.18) | Q(1) = 0.22; p = 0.64 | 12.69 (6.21, 19.18) | Q(1) = 1.83; p = 0.18 | 19.31 (8.11, 30.51) | Q(1) = 0.56; p = 0.46 | 80.69 (69.49, 91.89) | Q(1) = 0.56; p = 0.46 |
| | >18% participants | 48.10 (25.45, 70.74) | | 25.01 (8.39, 41.64) | | 27.19 (9.77, 44.60) | | 72.81 (55.40, 90.23) | |
| CKD | < = 19% participants | 66.11 (38.76, 93.46) | Q(1) = 2.86; p = 0.09 | 20.48 (9.50, 31.47) | Q(1) = 0.01; p = 0.92 | 25.78 (17.18, 34.38) | Q(1) = 0.78; p = 0.38 | 74.22 (65.62, 82.82) | Q(1) = 0.78; p = 0.38 |
| | >19% participants | 31.94 (28.37, 74.24) | | 13.40 (-1.51, 28.31) | | 15.25 (-6.53, 37.03) | | 84.75 (62.97, 106.53) | |
| Obesity | < = 19% participants | 45.59 (22.30, 68.89) | Q(1) = 0.10; p = 0.75 | 17.48 (5.20, 29.75) | Q(1) = 0.56; p = 0.45 | 17.86 (6.37, 29.35) | Q(1) = 0.32; p = 0.57 | 82.14 (70.65, 93.63) | Q(1) = 0.32; p = 0.57 |
| | >19% participants | 39.46 (9.69, 69.22) | | 15.98 (-10.15, 42.12) | | 26.21 (-0.58, 53.00) | | 73.79 (47.00, 100.58) | |

¥ Pooled estimate; * Test for the group differences; CI Confidence Interval; CKD Chronic Kidney Disease;

An analysis of subgroups by the proportion of hypertension (HTN) patients shows that the pooled estimate of ICU admission rate was 22% among the group of patients with more than 42% HTN patients and 12% among the group of patients with less than or equal to 42% HTN patients (Table 4). Both groups of participants, the group with more than 18% diabetes patients and the group with less than or equal to 18% diabetes patients, had ICU admission rates respectively 25% and 13%. Moreover, the group with more than 19% obesity patients had a 16% ICU admission rate, and the group with less than or equal to 19% obesity patients had a 17% ICU admission rate. Additionally, the pooled ICU admission rate for the group with at least 19% chronic kidney disease (CKD) patients was 13%, and for the group with less than or equal to 19% CKD patients, it was 20%. The ICU admission rates were not significantly different between the two groups for all co-morbidities considered.

The pooled estimate of the mortality rate among COVID-19 and multiple myeloma patients was 23% for the group with over 42% of patients with hypertension, compared to 21% for the group with 42% or fewer patients with hypertension (see Table 4). The estimated mortality rate was 27% for the group with over 18% of patients with diabetes, versus 19% for the group with 18% or fewer patients with diabetes. The pooled estimate for the mortality rate among patients with more than 19% obesity was 26%, while it was 18% for the group with 19% or fewer obese patients. The pooled mortality rate for groups with at least 19% chronic kidney disease (CKD) was 15%, and for those with 19% or fewer CKD patients, it was 26%. The mortality rates were not significantly different between the two groups for all co-morbidities considered.

The pooled estimate of the survival rate among COVID-19 and multiple myeloma patients was found to be 77% for the group with more than 42% HTN patients and 79% for the group with less than or equal to 42% HTN patients (Table 4). The estimated survival rate for the group with more than 18% diabetes patients was 73%, and for the group with less than or equal to 18% diabetes patients, it was 81% (Table 4). The pooled estimate of the survival rate among patients with at least 19% obesity was 74%, significantly lower than the survival rate of 82% for the group with less than or equal to 19% obesity patients. The pooled survival rate for the group with at least 19% chronic kidney disease (CKD) patients and the group with less than or equal to 19% CKD patients was 85% and 74%, respectively. There was no significant difference between the two groups for all co-morbidities considered.

## Meta-regression analysis

The meta-regression analysis indicates a significant positive relationship between the age of the participants and mortality rate and a negative association with survival rate (Table 5). The mortality rate increased by 15% for the group of participants whose median age was greater than 68 years compared to the groups of patients with a median age of less than or equal to 68 years. Similarly, among the co-morbidities obesity had a significant positive association with ICU admission rate, mortality rate, and a negative correlation with survival rate (Table 5). The ICU admission rate increased by 20% for the group of patients with at least 19% obesity compared to the group with less than or equal to 19% obesity. The mortality rate increased by 33% for the group of participants with at least 19% obesity compared to the group with less than or equal to 19% obesity. However, the survival rate decreased by 33% for the group of participants with at least 19% obesity compared to the group with less than or equal to 19% obesity.

## Discussion

The emergence of COVID-19 has posed significant challenges to healthcare systems globally, particularly in managing vulnerable populations such as patients with multiple myeloma

**Table 5. Meta-regression coefficients for study characteristics and patients' co-morbidities.**

| Factor | Labels | Hospitalization rate | | ICU admission rate | | Mortality rate | | Survival rate | |
|---|---|---|---|---|---|---|---|---|---|
| | | Coeff. (95% CI) | p-value | Coeff. (95% CI) | p-value | Coeff. (95% CI) | p-value | Coeff. (95% CI) | p-value |
| Participant and Study characteristics | | | | | | | | | |
| Study type | Case-series study | Ref. | | Ref. | | Ref. | | Ref. | |
| | Comparative cohort study | -38.24 (-80.08, 3.60) | 0.07 | -7.04 (-26.14, 12.06) | 0.47 | -22.11 (-45.93, 1.71) | 0.07 | 22.11 (-1.72, 45.93) | 0.07 |
| | Descriptove cohort study | -20.00 (-60.39, 20.28) | 0.33 | -3.02 (-21.75, 15.71) | 0.75 | -11.97 (-35.17, 11.22) | 0.31 | 11.97 (-11.22, 35.17) | 0.31 |
| Sample size | < = 129% participants | Ref. | 0.85 | Ref. | | Ref. | | Ref. | |
| | > 129% participants | 2.988 (-27.52, 33.50) | | -5.463 (-13.34, 2.41) | 0.17 | -4.83 (-23.14, 13.47) | 0.61 | 4.83 (-13.48, 23.14) | 0.61 |
| Median age | < = 68% participants | Ref. | 0.06 | Ref. | | Ref. | | Ref. | |
| | >68% participants | 24.771 (-0.77, 50.31) | | 4.297 (-2.46, 11.05) | 0.21 | 15.13 (0.51, 29.75) | 0.04 | -15.13 (-29.75, -0.51) | 0.04 |
| Proportion of men | < = 57% participants | Ref. | 0.52 | Ref. | | Ref. | | Ref. | |
| | >57% participants | 9.814 (-20.39, 40.01) | | 0.714 (-7.79, 9.22) | 0.87 | 9.48 (-8.71, 27.67) | 0.31 | -9.48 (-27.67, 8.71) | 0.31 |
| Participants co-morbidities | | | | | | | | | |
| Hypertension | < = 42% participants | Ref. | 0.71 | Ref. | 0.28 | Ref. | 0.35 | Ref. | 0.35 |
| | >42% participants | 7.87 (-32.88, 48.62) | | 4.95 (-4.02, 13.92) | | 8.92 (-9.76, 27.61) | | -8.92 (-27.61, 9.76) | |
| Diabetes | < = 18% participants | Ref. | 0.51 | Ref. | 0.89 | Ref. | 0.87 | Ref. | 0.87 |
| | >18% participants | -14.75 (-59.62, 30.11) | | 0.75 (-10.11, 11.61) | | -2.01 (-25.39, 21.37) | | 2.01 (-21.37, 25.39) | |
| CKD | < = 19% participants | Ref. | 0.33 | Ref. | 0.78 | Ref. | 0.97 | Ref. | 0.97 |
| | >19% participants | -30.46 (-90.76, 29.83) | | 2.50 (-15.33, 20.34) | | -.70 (-40.75, 39.36) | | .70 (-39.36, 40.75) | |
| Obesity | < = 19% participants | Ref. | 0.67 | Ref. | <0.001 | Ref. | <0.001 | Ref. | <0.001 |
| | >19% participants | 10.91 (-40.76, 62.58) | | 19.85 (7.35, 32.35) | | 33.36 (21.14, 45.59) | | -33.36 (-45.59, -21.14) | |

Coeff. coefficients; CI Confidence Interval; Ref. Reference group, CKD Chronic Kidney Disease

(MM) [28, 29]. This systematic review and meta-analysis provide comprehensive insights into the risk assessment and clinical implications of COVID-19 in MM patients.

The pooled estimates reveal concerning trends regarding the impact of COVID-19 on MM patients. The high hospitalization rate (53%) underscores the severity of illness experienced by

this group of patients, with a significant proportion requiring acute medical care [30]. Even though, individual study centers or hospitals show different rates due to different regional and patients' characteristics, most of the studies support our findings [4, 5, 11, 25]. Several studies observed that the majority of MM patients experienced critical/severe infections and health complications, necessitating hospitalization for a significant proportion [1, 7, 12, 23]. Conversely, in Israel during the pandemic, patients with hematological malignancies and COVID-19 exhibited a relatively lower hospitalization rate (32%) [9]. Furthermore, the substantial ICU admission rate (15.26%) emphasizes the critical nature of COVID-19 infections in MM patients, often requiring intensive interventions and specialized care [4, 31, 32]. A case series study in Brazil also noted increased hospitalizations, ventilatory support needs, and ICU admissions among this population [7]. Additionally, that study found that cross infection of COVID-19 and MM leads to frequent and significant complications at diagnosis and throughout treatment phases [7, 32, 33].

The pooled mortality rate (22%) and survival rate (78%) signify the heightened vulnerability of MM patients to adverse outcomes following COVID-19 infection. The COVID-19 pandemic's significant impact on MM patients was anticipated early on, given that viral infections, particularly respiratory ones, are common among this patient group [34]. Additionally, another research indicates that MM patients have mortality rates 50% higher than non-cancer patients when infected with COVID-19 [12]. Nevertheless, throughout the pandemic, the mortality rate for COVID-19 in patients with cancer has been reported to range between 11% and 28%, with rates as high as 37% for those specifically with hematologic malignancies [35–37]. While the majority of COVID-19 cases globally result in a mortality rate of 24%, the pooled estimated rate of 22% found in this study aligns closely with previous research (ranging from 10% to 39%), indicating a higher mortality rate compared to the general population and patients with other types of cancer [22, 25, 29]. However, one study has also found the fertility rate to be more than double, around 54%, among COVID-19 patients with MM [31]. We recognize that the observed variances in outcomes especially mortality rates across various countries and healthcare systems could be influenced by local epidemiological factors, hospital admission patterns, resource allocation, and potential disparities in the progression of medical intervention [22].

Subgroup analyses revealed several factors influencing the observed heterogeneity in outcomes among MM patients with COVID-19. Participants' age emerged as a significant predictor of mortality, with older MM patients exhibiting a higher risk of adverse outcomes. Furthermore, comorbidities such as obesity were associated with an increased risk of ICU admission and mortality. These findings are supported by recent literature, which collectively emphasizes the significant impact of age and obesity of MM patients on COVID-19 outcomes, such as increased rates of ICU admission, mortality, and decreased survival rates [9, 22]. The evidence emphasizes the importance of tailored management approaches for at-risk patient cohorts, encompassing individuals with advanced age, obesity, and comorbidities.While this study provides valuable insights into the risk assessment and clinical implications of COVID-19 in MM patients, several limitations should be acknowledged. Firstly, one of the primary limitations of this study was the insufficient availability of detailed and consistent data on risk factors across the included studies. Many of the original studies did not report specific risk factors associated with COVID-19 outcomes in multiple myeloma patients, or the reported data were not standardized or comprehensive enough to facilitate a meta-analysis on risk factors. Future studies should aim to provide more standardized and detailed reporting of risk factors associated with COVID-19 outcomes in multiple myeloma patients. Secondly, the inherent heterogeneity across included studies may limit the generalizability of findings, and potential publication bias cannot be entirely excluded despite rigorous methodological assessment.

Future research should consider strategies to address the inherent heterogeneity across studies, such as subgroup analyses based on study design, patient demographics, and disease characteristics. Thirdly, variations in diagnostic criteria, treatment protocols, and healthcare infrastructure across different settings may introduce additional sources of bias. Fourthly, some selected studies have very small sample sizes, which may reduce the precision of our estimates and introduce bias. Lastly, we were only able to include a small number of studies from specific countries in the final analysis, indicating that generalization of findings may not be appropriate for the entire world. In conclusion, this systematic review and meta-analysis underscore the significant impact of COVID-19 on MM patients, highlighting the need for targeted interventions and enhanced supportive care strategies. Further research is warranted to elucidate the underlying mechanisms driving adverse outcomes in this population and to inform evidence-based approaches for optimizing clinical management during the pandemic. By addressing these knowledge gaps, healthcare providers can better safeguard the health and well-being of MM patients in the face of evolving public health challenges.

## Supporting information

**S1 Table. PRISMA checklist.**
(DOCX)

**S2 Table. Search strategies and search results.**
(DOCX)

**S3 Table. JBI critical appraisal quality score.**
(DOCX)

**S1 File. Forest plots from subgroup analyses.**
(DOCX)

**S1 Protocol.**
(PDF)

## Author Contributions

**Conceptualization:** Sultan Mahmud, Md Mohsin.

**Data curation:** Sultan Mahmud, Md. Faruk Hossain.

**Formal analysis:** Sultan Mahmud, Md. Faruk Hossain.

**Investigation:** Sultan Mahmud.

**Methodology:** Sultan Mahmud.

**Project administration:** Sultan Mahmud.

**Resources:** Sultan Mahmud.

**Software:** Sultan Mahmud.

**Supervision:** Sultan Mahmud.

**Validation:** Sultan Mahmud, Md. Faruk Hossain, Abdul Muyeed, Md. Ashraful Haque, Harun Mazumder.

**Visualization:** Sultan Mahmud, Md. Faruk Hossain, Abdul Muyeed, Md. Ashraful Haque, Harun Mazumder.

**Writing – original draft:** Sultan Mahmud, Md. Faruk Hossain, Abdul Muyeed, Shaila Nazneen, Md. Ashraful Haque, Harun Mazumder, Md Mohsin.

**Writing – review & editing:** Sultan Mahmud, Md. Faruk Hossain, Abdul Muyeed, Shaila Nazneen, Md. Ashraful Haque, Harun Mazumder, Md Mohsin.

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
