## [Decision Letter · Decision Letter 0]

7 Jun 2024

PONE-D-24-17495Risk Assessment and Clinical Implications of COVID-19 in Multiple Myeloma Patients: A Systematic Review and Meta-AnalysisPLOS ONE

Dear Dr. Mahmud,

Thank you for submitting your manuscript to PLOS ONE. After careful consideration, we feel that it has merit but does not fully meet PLOS ONE’s publication criteria as it currently stands. Therefore, we invite you to submit a revised version of the manuscript that addresses the points raised during the review process.

The manuscript has undergone thorough review, and the feedback from the reviewers is enclosed with this letter. Two specialist reviewers in the field have identified several critical areas requiring major revisions. Overall, while the study addresses an important topic and presents valuable data, there are significant issues related to the clarity, methodological rigor, and completeness of the presented information. The study's quality and robustness could be greatly improved by providing clearer methodological details, enhancing the presentation of figures, and ensuring comprehensive data analysis, including sensitivity analyses and risk factor meta-analyses. Additionally, the manuscript would benefit from a more consistent use of terminology and thorough validation of key experimental components. Addressing these issues will enhance the study’s credibility and impact. Editors would be happy to consider the revised version for publication. When revising, address all points raised and outline every change made, or provide a suitable rebuttal if you disagree with any comments.

Best,

Dr. Abhinava K. Mishra

Academic Editor

We look forward to receiving your revised manuscript.

Kind regards,

Abhinava Kumar Mishra, PhD

Academic Editor

PLOS ONE

Journal Requirements:

2. Please amend the manuscript submission data (via Edit Submission) to include author Dr. Md Faruk Hossain.

3. Please amend your authorship list in your manuscript file to include author Dr. Sultan Faruk Mahmud.

Reviewers' comments:

Reviewer's Responses to Questions

**Comments to the Author**

1. Is the manuscript technically sound, and do the data support the conclusions?

Reviewer #1: Partly

Reviewer #2: Yes

2. Has the statistical analysis been performed appropriately and rigorously? 

Reviewer #1: Yes

Reviewer #2: Yes

3. Have the authors made all data underlying the findings in their manuscript fully available?

Reviewer #1: Yes

Reviewer #2: Yes

4. Is the manuscript presented in an intelligible fashion and written in standard English?

Reviewer #1: Yes

Reviewer #2: Yes

5. Review Comments to the Author

Reviewer #1: Thank you for this important work. Please find my suggestions below:

Why are there up to three references for PRISMA? How do the first two constitute PRISMA?

Can the authors briefly describe the JBI scoring under the quality assessment section?

Consider doing a leave-one-out sensitivity analysis to report the influence of each study on the pooled prevalence rates for your major outcomes.

Table 3 can be taken to the supplementary file.

The authors classified the available included studies into three: case series, retrospective cohort studies, and “retrospective observational studies” Can the authors define what they mean by “retrospective observational studies” since even cohort studies are also observational and can be retrospective (as in much of your included studies) or prospective, just as case series are also a type of observational studies.

Please, note that Egger’s test figures are missing from the main manuscript text and supporting files. Please, reference each figure when you include them in your revision.

Why have the authors not considered doing a meta-analysis on the risk factors for each of their four main outcomes (COVID-19-related hospitalization rate, ICU admission rate, mortality rate, and survival rate), especially given the fact that you stated your study’s title as “risk assessment”? Aren’t there data for such analysis from the original studies?

I look forward to reading the revised version of this manuscript

Reviewer #2: A great manuscript. Very comprehensive and most of analysis needed for a systematic review and meta analyses conducted well. Will possibly attract a lot of citations with the wealth of information that is being presented in this article.

6. PLOS authors have the option to publish the peer review history of their article (what does this mean?). If published, this will include your full peer review and any attached files.

Reviewer #1: **Yes: **Dr Sahabi Kabir Sulaiman

Reviewer #2: **Yes: **NAVIN KUMAR DEVARAJ

---

## [Author Response · Author response to Decision Letter 0]

24 Jun 2024

Response to Reviewer #1: 

Reviewer point #1: Why are there up to three references for PRISMA? How do the first two constitute PRISMA?

Author response #1: 

Thank you for catching this. We have removed these irrelevant citations from the revised manuscripts (page #2). 

Reviewer point #2: Can the authors briefly describe the JBI scoring under the quality assessment section?

Author response #2: 

Thank you for your valuable suggestion. We have added more description of the JBI scoring in both the Methods section (pages 3-4) and the Results section (page 8). During the update, we noticed some discrepancies in the JBI checklist. Therefore, we rescored all the studies using the most appropriate and updated checklist (see Supporting Information S3.). 

Reviewer point #3: Consider doing a leave-one-out sensitivity analysis to report the influence of each study on the pooled prevalence rates for your major outcomes.

Author response #3: 

Thank you for your insightful suggestion. We appreciate your recommendation to perform a leave-one-out sensitivity analysis to assess the influence of each study on the pooled prevalence rates for our major outcomes. This analysis will provide valuable insights into the robustness and stability of our findings.

We have conducted the leave-one-out sensitivity analysis as suggested. Both the Method (Page # 3) and Result section (Page# 9, Figure 4) has been updated accordingly. 

Reviewer point #4: Table 3 can be taken to the supplementary file.

Author response #4: 

Yes, we have added Table 3 as suppliementary information (S3 file). 

Reviewer point #5: The authors classified the available included studies into three: case series, retrospective cohort studies, and “retrospective observational studies” Can the authors define what they mean by “retrospective observational studies” since even cohort studies are also observational and can be retrospective (as in much of your included studies) or prospective, just as case series are also a type of observational studies.

Author response #5:

Thank you for your insightful comment. Yes, we completely agree with you the terminology is confusing as cohort studies are also observational by nature. To avoid confusion, we have revised our terminology in the revised manuscript to: 

• Case Series 

• Comparative Cohort Studies

• Descriptive Cohort Studies. 

We have also added clear definitions of each term in the Method section (Page#4-5) as follows:

• Case Series: Studies that present descriptive analysis of cases with a common characteristic, lacking a comparative group.

• Comparative Cohort Studies: These studies identify a cohort (group) of individuals who share a common characteristic or exposure in the past and then look back to compare outcomes between subgroups within this cohort. They typically include a comparison group and allow for some measure of association between exposure and outcome.

• Descriptive Cohort Studies: These studies analyze existing data without the formal structure of a cohort study. They often analyze data from medical records or databases to identify patterns, outcomes, and associations. Descriptive cohort studies do not involve comparing outcomes between different subgroups within the cohort.

Please note that, based on the new categorization, all analyses by study type have been updated. 

Reviewer point #6: Please, note that Egger’s test figures are missing from the main manuscript text and supporting files. Please, reference each figure when you include them in your revision.

Author response # 6: 

We have added a Table (Table 3) in revised manuscript (Page# 11), presenting the Egger test results. 

Reviewer point #7: Why have the authors not considered doing a meta-analysis on the risk factors for each of their four main outcomes (COVID-19-related hospitalization rate, ICU admission rate, mortality rate, and survival rate), especially given the fact that you stated your study’s title as “risk assessment”? Aren’t there data for such analysis from the original studies?

Author response #7: 

Thank you for your feedback. We appreciate the suggestion to conduct a meta-analysis on the risk factors for each of our four main outcomes (COVID-19-related hospitalization rate, ICU admission rate, mortality rate, and survival rate).

While we agree that such an analysis would be valuable, our primary limitation was the lack of detailed data on risk factors across the included studies. Most of the original studies did not provide sufficient information on specific risk factors. However, we have conducted subgroup analyses for study-level characteristics such as median age, proportion of men, and patients' comorbidities (hypertension, diabetes, obesity), identifying factors that may influence the outcomes of hospitalization rates, ICU admissions, mortality rates, and survival rates.

In our manuscript, we aimed to highlight the available data on these outcomes and acknowledged the limitations posed by the variability and incompleteness of risk factor reporting (Page #15). We believe that addressing this gap is crucial for future research, and we have included a recommendation for more standardized and detailed reporting of risk factors in original studies in our discussion section (Page #15).

Response to Reviewer #2: 

Reviewer point #1: A great manuscript. Very comprehensive and most of analysis needed for a systematic review and meta analyses conducted well. Will possibly attract a lot of citations with the wealth of information that is being presented in this article.

Author response #1: 

Thank you very much for your positive feedback. We are delighted to hear that you found our manuscript comprehensive and well-conducted. We appreciate your kind words regarding the potential impact and value of our work. 

Additional requirements:

2. Please amend the manuscript submission data (via Edit Submission) to include author Dr. Md Faruk Hossain.

3. Please amend your authorship list in your manuscript file to include author Dr. Sultan Faruk Mahmud.

Author response: 

Thank you very much for your suggestions. We have ensured that our manuscript meets PLOS ONE's style requirements, including file naming, as per the guidelines provided. The author list in the manuscript was correct; however, we have amended the manuscript submission data to fix the authorlist.

---

## [Decision Letter · Decision Letter 1]

23 Jul 2024

Risk Assessment and Clinical Implications of COVID-19 in Multiple Myeloma Patients: A Systematic Review and Meta-Analysis

PONE-D-24-17495R1

Dear Dr. Mahmud,

We’re pleased to inform you that your manuscript has been judged scientifically suitable for publication and will be formally accepted for publication once it meets all outstanding technical requirements.

Kind regards,

Abhinava Kumar Mishra, PhD

Academic Editor

PLOS ONE

Additional Editor Comments (optional):

Reviewers' comments:

Reviewer's Responses to Questions

**Comments to the Author**

1. If the authors have adequately addressed your comments raised in a previous round of review and you feel that this manuscript is now acceptable for publication, you may indicate that here to bypass the “Comments to the Author” section, enter your conflict of interest statement in the “Confidential to Editor” section, and submit your "Accept" recommendation.

Reviewer #1: All comments have been addressed

2. Is the manuscript technically sound, and do the data support the conclusions?

Reviewer #1: Yes

3. Has the statistical analysis been performed appropriately and rigorously? 

Reviewer #1: Yes

4. Have the authors made all data underlying the findings in their manuscript fully available?

Reviewer #1: Yes

5. Is the manuscript presented in an intelligible fashion and written in standard English?

Reviewer #1: Yes

6. Review Comments to the Author

Reviewer #1: Thank you for revising this important manuscript.

After going through the revision, I can see that the manuscript has significantly improved.

Looking forward to seeing the published version of this work.

7. PLOS authors have the option to publish the peer review history of their article (what does this mean?). If published, this will include your full peer review and any attached files.

Reviewer #1: **Yes: **Dr Sahabi Kabir Sulaiman

---

## [Editor Report · Acceptance letter]

25 Jul 2024

PONE-D-24-17495R1 

PLOS ONE

Dear Dr. Mahmud, 

I'm pleased to inform you that your manuscript has been deemed suitable for publication in PLOS ONE. Congratulations! Your manuscript is now being handed over to our production team.

Kind regards, 

on behalf of

Dr. Abhinava Kumar Mishra 

Academic Editor

PLOS ONE